# Corrosion Protection Mechanism Study of Nitrite-Modified CaAl-LDH in Epoxy Coatings

**Junhao Xue** [1,2]**, Jingjing Wang** [1,2,*]**, Yanhui Cao** [1,2,*]**, Xinyue Zhang** [1,2]**, Lili Zhang** [1,2]**, Kaifeng Chen** [1,2] **and Congshu Huang** [1,2]

[1] Luoyang Ship Material Research Institute, Xiamen 361101, China; dnxjh666@163.com (J.X.); ohtlys@sina.cn (X.Z.); xunmengren_82@126.com (L.Z.); xm725ckf@163.com (K.C.); hcs05@126.com (C.H.)

[2] Science and Technology on Marine Corrosion and Protection Laboratory, Xiamen 361101, China

[*] Correspondence: jingjing811014@163.com (J.W.); yanhuicao@yeah.net (Y.C.)

**Abstract:** In this work, nitrite and molybdate-modified CaAl layered double hydroxide(CaAl-LDH) was first synthesized, and the corrosion protection mechanism of CaAl-LDH intercalated with nitrites in epoxy coatings was investigated. Scanning electronic microscopy (SEM) and the energy dispersive spectroscopy (EDS) was used to characterize the morphology and element composition of the synthesized powder. Fourier transform infrared spectroscopy (FTIR) was used to characterize the information of chemical composition, and X-ray diffraction (XRD) was used to analyze the structure. The SEM and XRD results indicated that the LDH structure was destroyed in the molybdate modification process, and $CaMoO_4$ precipitates were formed. Therefore, molybdates cannot be used to be loaded in CaAl-LDH interlayer space for synthesis of an active corrosion inhibition container. The nitrite release curve and the chloride concentration decreasing curve were measured to study the anion-exchange reaction by UV-Vis spectroscopy and a home-made Ag/AgCl probe, respectively. The corrosion protection effect of the CaAl-LDH loaded with nitrites towards the carbon steel was evaluated in 0.02 M NaCl solution by electrochemical impedance spectroscopy (EIS). Then the powder was added in the epoxy coating with 2% addition (weight vs. epoxy resin). The coating morphology and roughness were evaluated by SEM and laser microscopy, and the corrosion protection effect was investigated by EIS in an immersion period of 21 d. The fitted coating resistance of the sample with 2% LDH intercalated with nitrites was one order of magnitude higher than that with 2% LDH, and the latter one was two orders of magnitude higher than the blank sample. Local electrochemical impedance spectra (LEIS) was used to characterize the corrosion development process in micro-corrosion sites. The corrosion product of the scratched area after salt spray exposure was analyzed by EDS and Raman spectroscopy. The corrosion protection mechanism of the CaAl-LDH loaded with nitrites was proposed based on the above experimental results.

**Keywords:** layered double hydroxide; epoxy coating; corrosion protection mechanism

## 1. Introduction

Epoxy coatings have been used as effective coatings to protect metal substrate from serious corrosion. However, some micro-cracks, micropores, or defects can be produced during the curing process, which could act as the transportation path of water molecules and aggressive anions in the coatings. When the water molecules and aggressive chlorides reach the coating/substrate interface, corrosion will be initiated and further propagated if no inhibitors exist in this system [1]. Therefore, to solve this problem, researchers developed lots of methods to prevent corrosion initiation and propagation [2,3].

To enhance the permeation difficulty of water molecules and aggressive chlorides, researchers added 2D materials with high aspect ratio into the polymer matrix, such as graphene, basalt, hexagonal boronnitride, MXene, layered double hydroxides, etc. [4]. The poor dispersion and serious aggregation limited their application in coatings. Much

effort had been devoted to improve the dispersion state in the epoxy coatings. For example, Wen Sun et al. synthesized graphene/pernigraniline composites, and pernigraniline growing in graphene surface was able to prevent stacking of graphene and prolonged the diffusion path of aggressive anions in the coating matrix [5]. Siming Ren et al. used boron nitride (BN) monolayer to protect Cu substrate for long-term corrosion protection based on the effective physical barrier effect in corrosive medium [6]. Mingjun Cui et al. investigated the corrosion inhibition performance of epoxy coatings containing water-dispersible boron nitride sheets [4]. Yulong Wu et al. grew ZnAl-LDH on the surface of Mxene to inhibit its self-restacking and spontaneous aggregation [7,8]. In addition, Meng Zhang et al. prepared LDH loaded with molybdate on the surface of PDA-modified basalt, which could avoid stacking of the LDH platelet and also increased the interfacial compatibility between fillers and epoxy resin [1]. As reported by Seyyed Arash Haddadi et al., the $NO_3^-$ intercalated ZnAl-based layered double hydroxides were used to decorate graphene oxide to prevent the formation of agglomeration and poor dispersion of graphene oxide [9].

In addition, researchers developed various inhibitor nanocontainers and incorporated them in the epoxy coatings. The nanocontainers mainly include layered double hydroxides (LDHs), mesoporous silica, metal-organic frameworks (MOFs), halloysites, montmorillonite, etc. [10–16]. For example, Changhua Li et al. incorporated MXene@MgAl-LDH intercalated with $MoO_4^{2-}$ into epoxy coating with active corrosion inhibition due to the anion-exchange between $Cl^-$ and $MoO_4^{2-}$ [8]. Dashuai Yan et al. used mesoporous silica as the inhibitor container and loaded them with cysteine and iron polyacrylate, which were added into the bottom and top epoxy coating of a double-layered coating system, respectively. The coating exhibited an excellent corrosion protection effect [17]. Yanning Chen et al. developed MOF-decorated graphene oxide/MgAl-LDH coating through microstructural optimization with high anticorrosion performance [18]. Yanling Jia et al. included inhibitor L-histidine (L-His) in graphene/halloysite nanotubes and synthesized intelligent nanocontainers with pH responsive property, and applied them as a modifier in waterborne coatings. The driving force of the release of the intercalated inhibitor L-His is mainly resulting from the electrostatic adsorption, and the responsive release curve was different in acid and alkaline solutions [19]. I. Mohammadi et al. synthesized cerium loaded Na-montmorillonite (Na-MMT) and incorporated them in the epoxy coating. The active inhibition performance of the containers can be ascribed to the $Ce^{3+}$ release from the Ce-MMT [20]. Xuteng Xue et al. loaded $Na_2MoO_4$ and benzotrialoze (BTA) into halloysite nanotubes (HNTs) and obtained Cu-BTA-$Na_2MoO_4$-HNTs, and this composite demonstrated promising efficient corrosion protection with acid-response ability for the carbon steel [13].

Compared to other micro/nanocontainers, the most attractive characteristic of LDHs can be attributed to the anion-exchange capability [21]. This suggested that inhibitors can be released only on demand. When aggressive anions appear in the external environment, they can be adsorbed into the interlayer space, and correspondingly, the intercalated inhibitor anions can be released to heal the corrosion site and inhibit corrosion propagation. This controlled release of inhibitors depending on the changes of the environment is able to provide efficient corrosion protection. It should be noted that the anion-exchange reaction was influenced by a wide range of factors including pH, temperature, the size and charge of the loaded anions and the anions in surrounding environment, and the concentration of the anions [22]. Thus, the anion-exchange capability and kinetics could be adjusted by changing the above factors.

Yanning Chen et al. prepared ternary Mg-Al-La layered double hydroxides (LDHs) (ternary L/G-V coating) loaded with vanadate ($V_2O_7^{4-}$) doped with graphene oxide (GO)-doped. The effective inhibition effect of this L/G-V coating was attributed to the release of $V_2O_4^{7-}$ and the formation of $Mg_4(V_2O_7)_2$ on the anodic area. The La cations and vanadate anions play a synergistic role, and thus, have a dual self-healing effect [23]. They further developed MOF-decorated graphene oxide/MgAl-LDH on the micro-arc oxide coating on AZ31 magnesium alloy [18]. Tatsiana Shulha et al. intercalated two kinds of typical

inhibitor, including oxalates and vanadates into MgAl-LDH. The results demonstrated that LDH loaded with vanadates is much more effective for the active protection of magnesium than the LDH loaded with oxalates [24]. Mohammad Tabish et al. prepared CaAl-LDH intercalated with 2-mercaptobenzothiazole (MBT) and added them in epoxy coating with different quantities, and the obtained coating showed a self-healing effect to the carbon steel [25]. In recent publications, LDH was frequently used in combination with other species, including MOF, ZIF, GO, and MXene to obtain a better corrosion protection effect in coatings [8,26,27].

In our previous work, the improved corrosion protection effect of the coating with the addition of LDH was mainly ascribed to the physical barrier effect of incorporated LDH platelets, while the chloride trapping effect caused by the anion-exchange reaction made little contribution when no effective inhibitors were loaded in the LDH interlayer space [28]. In this work, we further worked on the corrosion mechanism study of LDH in epoxy coatings when effective traditional inhibitor anions were intercalated in the LDH gallery. CaAl-LDH intercalated with nitrites was synthesized firstly in this work and added into the epoxy coating. A wide range of characterization methods was used to characterize the LDH powder and the coating. Finally, the corresponding corrosion mechanism was proposed.

## 2. Experimental Section

### 2.1. Materials and Agents

The chemical agents including $Ca(NO_3)_2 \cdot 4H_2O$, $Al(NO_3)_3 \cdot 9H_2O$, $NaNO_3$, $NaCl$, $NaOH$, $NaNO_2$, and $Na_2MoO_4$ were ordered from Sinopharm Chemical Reagent Co., Ltd., (Beijing, China). All of them were analytical grade. The deionized water was used for solution preparation. The 6101 epoxy resin and the 2519 curing agent were provided by Shanghai Dekun Industrial Co., Ltd., (Shanghai, China).The specifications related to physical and chemical properties of them were listed in Tables 1 and 2. n-butanol was used as the solvent, which was provided by Baling Petroleum & Chemical Co., Ltd., (Yueyang, China). The Q235 carbon steel was purchased from Xiamen Qianfeng Mechanical Co., Ltd., (Xiamen, China). The defoaming agents 530, leveling agent 320, and dispersing agent 110 were provided by Bike Chemial Co., Ltd., (Tongling, China).

**Table 1.** The specifications related to main physical and chemical properties of 6101 epoxy resin.

| Items | 6101 Epoxy Resin |
|---|---|
| Epoxy Equivalent (g/eq) | 210–240 |
| Softening Point (°C) | 12–20 |
| Viscosity (25 °C mPa·s80% Xylene Solution) | 200–400 |
| Saponifiable Chlorine (%) | ≤0.3 |
| Inorganic Chlorine (ppm) | ≤180 |

**Table 2.** The specifications related to main physical and chemical properties of 2519 hardener.

| Items | 2519 Hardener |
|---|---|
| Viscosity (25 °C mPa·s) | 185 |
| Density (25 °C g/mL) | 1.01 |
| Flash Point (°C) | >100 |
| Amine Value (mg KOH/g) | 315 |

### 2.2. LDH Preparation and Modification

Co-precipitation method was used to prepare CaAl-LDH. The solution containing 0.5 M $Ca(NO_3)_2$ and 0.25 M $Al(NO_3)_3$ was added drop wisely into 2 M $NaNO_3$ and 3 M $NaOH$ with high dispersion rate at 65 °C. The slurry was then transferred into the hydrothermal stainless reactor and subjected to hydrothermal reaction in oven at 120 °C for 24 h. After washing by deionized water and ethanol, the obtained reactant was finally dried at 60 °C for 48 h and it was labeled as CaAl-LDH.

Then, 1 g CaAl-LDH was added into 100 mL 0.1 M NaNO$_2$ and the solution was stirred rigorously for 24 h in ambient atmosphere at room temperature. Then the solution was centrifuged at a rate of 8000 rpm for 5 min and the obtained solid was subjected to drying in oven for 48 h at 60 °C. The obtained sample was marked as CaAl-LDH-NO$_2^-$. For CaAl-LDH-MoO$_4$, 0.1 M Na$_2$MoO$_4$ was used to replace 0.1 M NaNO$_2$ instead, as described above.

### 2.3. LDH Characterization

The morphology of CaAl-LDH, CaAl-LDH-NO$_2^-$, and CaAl-LDH-MoO$_4^{2-}$ was observed by scanning electronic microscopy (SEM, ZEISS ULTRA55, Carl Zeiss AG Co., Ltd., Dresden, Germany). The elemental composition of these samples was detected by the coupled energy dispersive spectrometer (EDS, Carl Zeiss AG Co., Ltd., Dresden, Germany). Fourier transform infrared spectroscopy (FTIR, Nicolet IS10, Thermo Fisher Scientific Co., Ltd., Waltham, MA, USA) was adopted to characterize the chemical composition of the synthesized LDH samples, the scanning was from 400 to 4000 cm$^{-1}$, and the resolution was 4 cm$^{-1}$. X-ray diffraction (XRD, X'Pert PRO, Malvern PANalytical Co., Ltd., Almelo, the Netherlands) was used to analyze the structure of LDH, and it was scanned from 5 to 80° with a scanning rate of 10°/min.

The nitrite release curve was measured by UV-Vis spectroscopy (Cary 7000, Agilent, Santa Clara, CA, USA) at the wavelength of 354 nm based on the calibration curve. LDH-NO$_2^-$ was dispersed in 0.02 mol/L NaCl solutions with magnetic stirring at a concentration of 5 g/L, and 2 mL solution was taken at fixed intervals, which was then subjected to UV-Vis spectroscopy measurement. The home-made Ag/AgCl probe prepared by a constant current anodizing method was used to measure the changes of chloride concentration in this solution at certain intervals.

The corrosion protection ability of the synthesized LDH powders in 0.02 M NaCl solution towards carbon steel was evaluated by the electrochemical impedance spectroscopy (EIS) on an Autolab electrochemical workstation (Metrohm China Limited, Beijing, China). The scanned frequency was from 10$^5$–10$^{-2}$ Hz and the adopted sinusoidal disturbance was 10 mV. The EIS spectra were recorded every day in an immersion period up to 7 d.

### 2.4. Coating Preparation

Component A was prepared according to the following procedure: 4 g LDH-NO$_2^-$ was added in 100 mL n-butanol. The mixture was stirred violently for 20 min and was then ultrasonically dispersing for 30 min. Afterwards, the n-butanol was mixed with 200 g epoxy resin (E-44). In addition, 2 g defoaming agents 530, 2 g leveling agent 320, and 2 g dispersing agent 110 were also added into the above mixture. The mixture was subjected to rigorous dispersion with a dispersing rate of 1000 r/min. The curing agent 2519 was used as component B. Please note that component A and B were mixed with a ratio of 300:84 using a wooded stick. This mixture was applied on the carbon steel plate (70 mm × 150 mm) using a brush. After curing for 24 h, the mixture will be applied on the coated carbon steel plate again. The thickness of the coating was 120 ± 20 μm. For comparison, component A with and without 4 g LDH was also prepared.

### 2.5. Coating Characterization

The morphology of the prepared epoxy coating including topview and sideview was characterized by scanning electronic microscopy (SEM). The 3D morphology and the roughness were measured by laser microscopy (KEYENCE, VK-X150, Osaka, Japan) with a laser wavelength of 658 nm. The coating's corrosion protection ability was evaluated by EIS. The EIS was measured by Electrochemical Impedance Analyzer (Wuhan Kesite Instrument Co., Ltd., Wuhan, China). The scanned frequency was 10$^5$–10$^{-2}$ Hz, and the sinusoidal disturbance was set as 10 mV. The EIS spectra were measured every dayin a long immersion period of 21 d. The EIS data was fitted by Zview3.0 (Scribner Associates Inc., Southern Pines, NC, USA). In addition, the corrosion protection effect was investigated

by salt spray test as well for 7 d according to GB/T 1771-2007. Before the salt spray test, the coating sample was scratched artificially and the underlying bare metal substrate was exposed to the corrosive salt spray. After the salt spray test, the optical images of the samples were captured. The samples were washed by flowing water rigorously. Then the rust was removed from the scratched area and EDS results of the rust were obtained. In addition, DXR Raman Microscope (ThermoFisher, Waltham, MA, USA) was also used to characterize the composition of the rust to analyze the difference of different samples with an excitation laser wavelength of 633 nm. The localized electrochemical impedance spectroscopy (LEIS) was measured to evaluate the self-healing behavior of the samples on a scanning electrochemical probe measurement system (M470, Bio-Logic Science Instruments, Seyssinet-Pariset, France). The amplitude was 10 mV and the adopted frequency was 50 Hz. The scanning area is 4 mm × 4 mm. The corresponding spectra was recorded after exposing to 3.5 wt.% NaCl for 1 h, 24 h, and 48 h.

## 3. Results and Discussion

### 3.1. Characterization of the Synthesized LDH Powders

According to Figure 1a,b, the synthesized CaAl-LDH presented hexagonal shape with wide distributions of sizes. The size of most CaAl-LDH platelets was larger than 2 µm approximately. The obvious aggregations of CaAl-LDH also appeared as marked by the red circles in Figure 1b. The EDS results indicated that the CaAl molar ratio was 1.70, which was close to the theoretical value of 2.

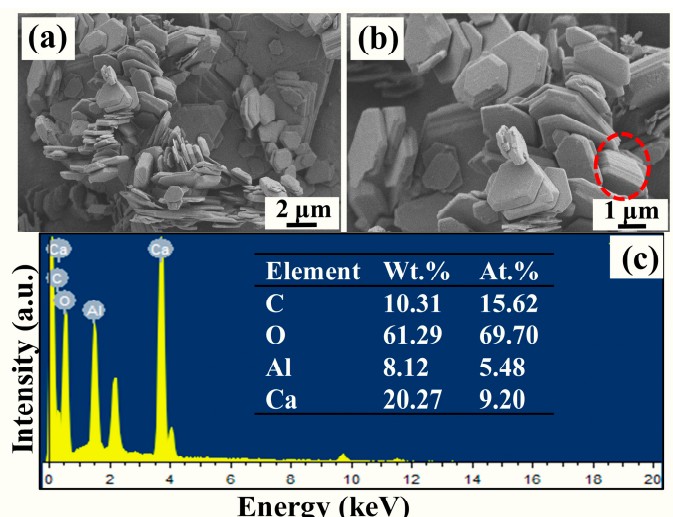

**Figure 1.** The SEM and EDS results of CaAl-LDH. (**a**) The morphology, (**b**) The morphology and (**c**) EDS composition.

The element distribution of CaAl-LDH is presented in Figure 2. It can be seen that the Ca, Al, and O distributed uniformly corresponding to the LDH platelets in Figure 2a, while the distribution of the C element was different from the other three elements as the detected substrate of the conductive adhesive tape was rich in the C element.

According to Figure 3a,c, the morphology of CaAl-LDH-$NO_2^-$ was similar to that of pristine CaAl-LDH without obvious changes. However, after modification by $Na_2MoO_4$, the obtained CaAl-LDH-$MoO_4^{2-}$ presented flower-like morphology, which was completely different from that of CaAl-LDH and CaAl-LDH-$NO_2^-$. It was probable that CaAl-LDH suffered from significant phase transformation during the immersion process in the $Na_2MoO_4$ solution. In addition, the EDS results of CaAl-LDH-$NO_2^-$ showed a CaAl mole ratio of 1.87, which was close to the theoretical value of 2. The N element was detected, verifying the existence of the nitrites. Based on the EDS result of CaAl-LDH-$MoO_4^{2-}$ in Figure 3f, a significant amount of the Mo element can be observed in the EDS spectrum, indicating the existence of $MoO_4^{2-}$. The Al element became negligible; this result suggested that the LDH

structure have been destroyed probably and some new substance was formed, probably during the modification process. Other characterizations were needed to further define the formed products.

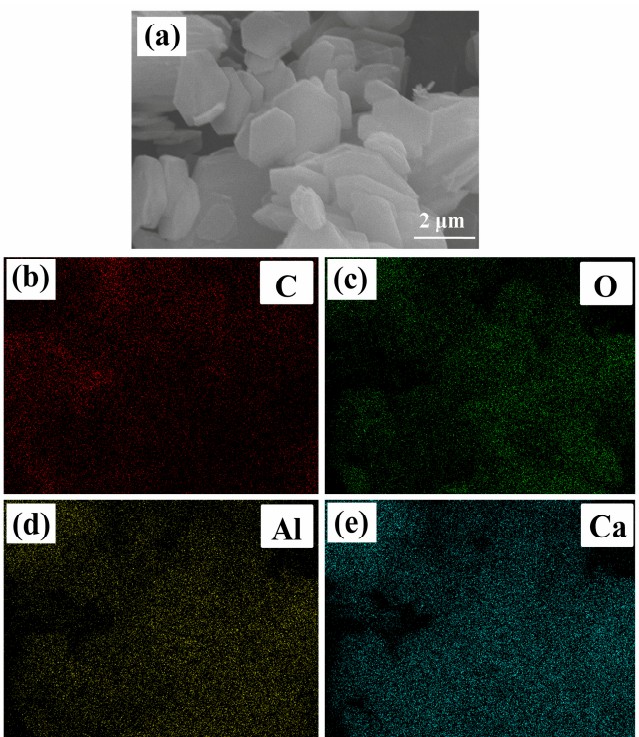

**Figure 2.** The element distribution of CaAl-LDH. (**a**) The morphology, (**b**) C element (**c**) O element (**d**) Al element and (**e**) Ca element.

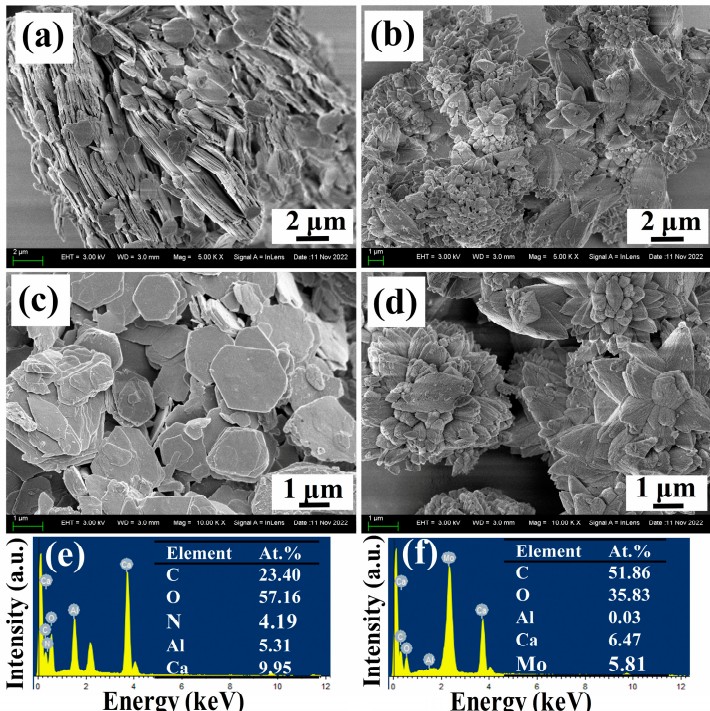

**Figure 3.** The morphology and element composition of CaAl-LDH-NO$_2^-$ and CaAl-LDH-MoO$_4^{2-}$. (**a**) The morphology of CaAl-LDH-NO$_2^-$, (**b**) The morphology of CaAl-LDH-MoO$_4^{2-}$, (**c**) The morphology of CaAl-LDH-NO$_2^-$, (**d**) The morphology of CaAl-LDH-MoO$_4^{2-}$, (**e**) The EDS composition of CaAl-LDH-NO$_2^-$, (**f**) The EDS composition of CaAl-LDH-MoO$_4^{2-}$.

Figure 4 shows the XRD results of CaAl-LDH, CaAl-LDH-NO$_2^-$, and CaAl-LDH-MoO$_4^{2-}$. It was clear from Figure 4a that the CaAl-LDH and CaAl-LDH-NO$_2^-$ present well-defined peaks of LDH, corresponding to (003) plane and (006) plane [29], while the sample of CaAl-LDH- MoO$_4^{2-}$ only presented peaks related to CaMoO$_4$ (PDF29-0351), and the peaks related to LDH disappeared completely. The peak of (003) plane of CaAl-LDH and CaAl-LDH-NO$_2^-$ occurred at 10.2 and 11.1, respectively. The corresponding interlayer distance was calculated to be 0.867 and 0.796 nm based on the Bragg's Law, respectively. According to Figure 4b, some peaks attributed to CaCO$_3$ could be observed in the XRD spectra of CaAl-LDH (PDF47-1743), indicating the formation of impurities in the LDH synthesis process. The intensity of CaAl-LDH-NO$_2^-$ decreased obviously in comparison with CaAl-LDH, suggesting the declined crystallinity of LDH during the modification process in NaNO$_2$ solution. The peaks related to CaMoO$_4$ can be defined clearly, corresponding to different planes of CaMoO$_4$, which have been marked in Figure 4b. It was concluded that the CaAl-LDH underwent structure transformation during immersion in the Na$_2$MoO$_4$ solution, CaAl-LDH may dissolve, and insoluble CaMoO$_4$ was formed, finally. This result was in good agreement with the aforementioned SEM and EDS results in Figure 3b,d,f. In the reported literature, Na$_2$MoO$_4$ was used to modify LDH such as ZnAl-LDH and MgAl-LDH frequently, and the obtained LDH usually presented excellent corrosion inhibiting properties [30–32]. In this work, Na$_2$MoO$_4$ was used to modify CaAl-LDH for the first time and, it can be found that the main product resulted to be CaMoO$_4$, demonstrating that MoO$_4^{2-}$ inhibitor anions cannot be used for the modification of CaAl-LDH. This newly reported phenomenon was discovered for the first time, and could provide useful instruction information for the researchers and engineers in the relevant field.

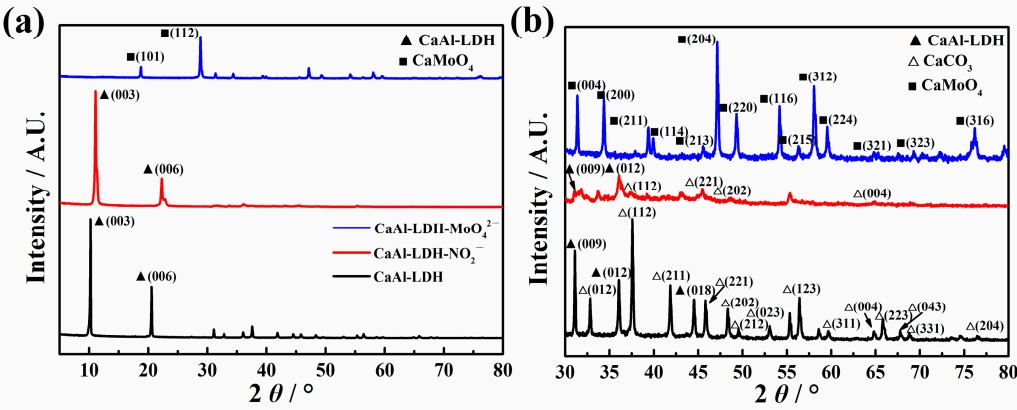

**Figure 4.** XRD results of different samples (**a**) 5–80°; (**b**) the magnified XRD spectrum from 30 to 80°.

The FTIR results of different samples CaAl-LDH, CaAl-LDH-NO$_2^-$, and CaAl-LDH-MoO$_4^{2-}$ are listed in Figure 5. The peaks around 530 and 420 cm$^{-1}$ in the FTIR spectrum of CaAl-LDH and CaAl-LDH-NO$_2^-$ can be ascribed to M-O and M-OH in the LDH lattice, respectively [25]. The peaks around 1635 cm$^{-1}$ in the three curves can be ascribed to bending stretching of H$_2$O molecules [32]. As for CaAl-LDH, the spectrum in the range of 1290–1420 cm$^{-1}$ was magnified, which was presented in Figure 5b. The peak at 1352 and 1384 cm$^{-1}$ can be attributed to the carbonates and nitrates, respectively [33]. The peak at 1270 cm$^{-1}$ at the curve of CaAl-LDH-NO$_2^-$ was due to the characteristic stretching of NO$_2^-$ [34]. The small peaks of CaAl-LDH-NO$_2^-$ attributed to nitrates and carbonates at 1383 and 1353 cm$^{-1}$ could also observed. The dramatic decrease of the peak around 1353 cm$^{-1}$ in comparison with that of CaAl-LDH was probably caused by the anion-exchange reaction between the carbonates in the LDH interlayer space and the nitrites in the surrounding environment during the nitrite modification process. The broad peak around 791 cm$^{-1}$ in the curve of CaAl-LDH-MoO$_4^{2-}$ was due to Mo-O stretching vibration in MoO$_4^{2-}$ tetrahedrons [30].

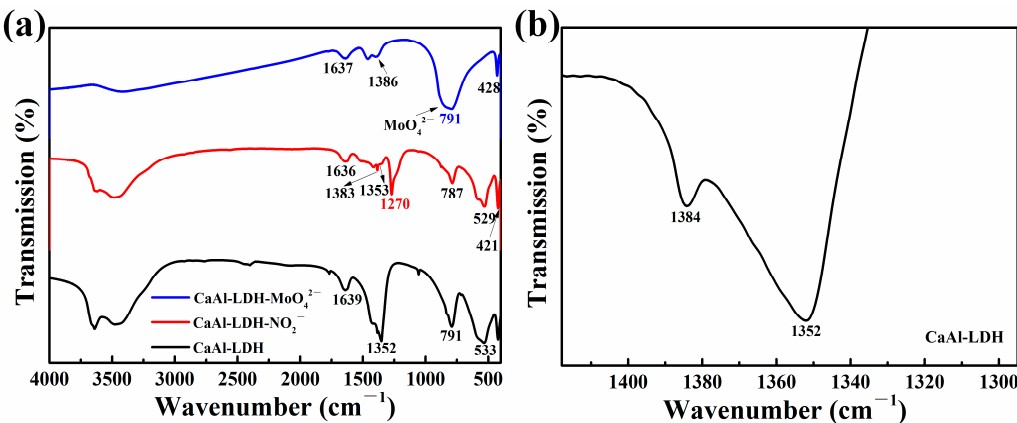

**Figure 5.** (**a**) FTIR results of CaAl-LDH, CaAl-LDH-NO$_2^-$ and CaAl-LDH-MoO$_4^{2-}$. (**b**) The magnified FTIR results of CaAl-LDH.

*3.2. Corrosion Protection Ability of the Synthesized LDH Powders in 0.02 M NaCl Solution*

To evaluate the corrosion protection ability of LDH-NO$_2^-$ on the carbon steel, it was incorporated into 0.02 mol/L NaCl at a concentration of 5 g/L. For comparison, the EIS measurements of carbon steel in blank NaCl solution and in NaCl solution with 5 g/L LDH were also performed. The EIS results were recorded after immersion for 2 h in an immersion duration of 7 d, since open circuit potential(OCP) reached a steady state only after 0.5 h [34,35]. Both the Nyquist and Bode plots are presented in Figure 6. According to Figure 6(a1,b1,a2,b2), the impedance arc of the sample in blank NaCl solution and in NaCl solution with 5 g/L LDH decreased remarkably after immersion of 1 d in comparison with that of 2 h. The impedance presented a decreasing trend in this immersion period of 7 d in spite of some small fluctuations. According to Figure 6(b1,b2), the impedance values at 0.01 Hz of the carbon steel in NaCl solution with 5 g/L LDH were more than one magnitude larger than that of the carbon steel in blank NaCl solution, which may be due to the chloride adsorption effect based on the anion exchange characteristic and the enhanced physical barrier effect caused by the plate-like morphology of LDH. In addition, the phase angle peak in Figure 6(c2) was much broader than that in Figure 6(c3), indicating certain corrosion protection of LDH powder in NaCl solution towards the carbon steel. When LDH-NO$_2^-$ was added in the 0.02 M NaCl solution (Figure 6(a3,b3)), the obtained impedance arc became much larger than that in the blank NaCl solution and the NaCl solution with addition of 5 g/L LDH. The impedance values at the low frequency of 0.01 Hz was one magnitude larger than that of the carbon steel immersed in NaCl solution with the addition of 5 g/L LDH, indicating an enhanced corrosion protection effect of LDH-NO$_2^-$ compared with that of LDH. Accordingly, the phase angle peak in Figure 6(c3) was much broader than that in Figure 6(c2). The equivalent circuit containing one time constant in Figure 6d was adopted to fit the EIS data, where $R_s$ represented solution resistance, the $R_{ct}$ meant the charger transfer resistance related to the electrochemical corrosion reaction, and the $CPE_{dl}$ meant the double layer capacitance, which was used here to replace an ideal capacitor due to the non-homogeneity of the carbon steel surface in this system [35,36]. The impedance of $CPE$ can be calculated by the following expression [35,37]:

$$Z_{CPE} = Y^{-1}(jw)^{-n} \tag{1}$$

where $Y$ ($\Omega^{-1}$ cm$^{-2}$ s$^n$) is a proportional factor, s, $j$ is the imaginary number, $\omega$ is the frequency in radians per second, $n$ is a factor ranging from 0 to 1. The higher the value of $n$ is, the electrode surface is more uniform and denser. The detailed parameters obtained through fitting are listed in Table 3. According to Figure 6e, the $R_{ct}$ values of the sample immersed in NaCl solution with 5 g/L LDH-NO$_2^-$ indicated a rising trend with the increased immersed time and at least one magnitude higher than other samples; in contrast, the $R_{ct}$ values of the sample immersed in blank NaCl solution and in NaCl solution decreased continuously. In addition, the $Y_{dl}$ values of various samples presented a different

trend in comparison with that of $R_{ct}$ values. According to the literature, $Y_{dl}$ measures the numbers of the electrochemically active sites in the interface between the coating and substrate [28,38]. The $Y_{dl}$ of the carbon steel in blank NaCl solution presented the largest values and the corresponding value of the carbon steel in NaCl solution with 5 g/L LDH was lower; however, this value was still larger than that of carbon steel in NaCl solution with 5 g/L LDH-NO$_2$$^-$. This result indicated that corrosion can be effectively prevented in the presence of LDH-NO$_2$$^-$ in this system. In Figure S1, after the addition of LDH-MoO$_4$$^{2-}$, the corrosion resistance of the carbon steel was a little bit larger than that of the carbon steel in blank solution, and much smaller than that of the sample with LDH and LDH-NO$_2$$^-$, which is probably due to the formation of CaMoO$_4$, as verified in the SEM results in Figure 3 and the XRD results in Figure 4. Therefore, this sample was not added in the epoxy coating for further corrosion test. However, the above result could provide instruction significance in future LDH modification, and some specific inhibitor was not suitable for modification of certain LDH due to the possible reactions.

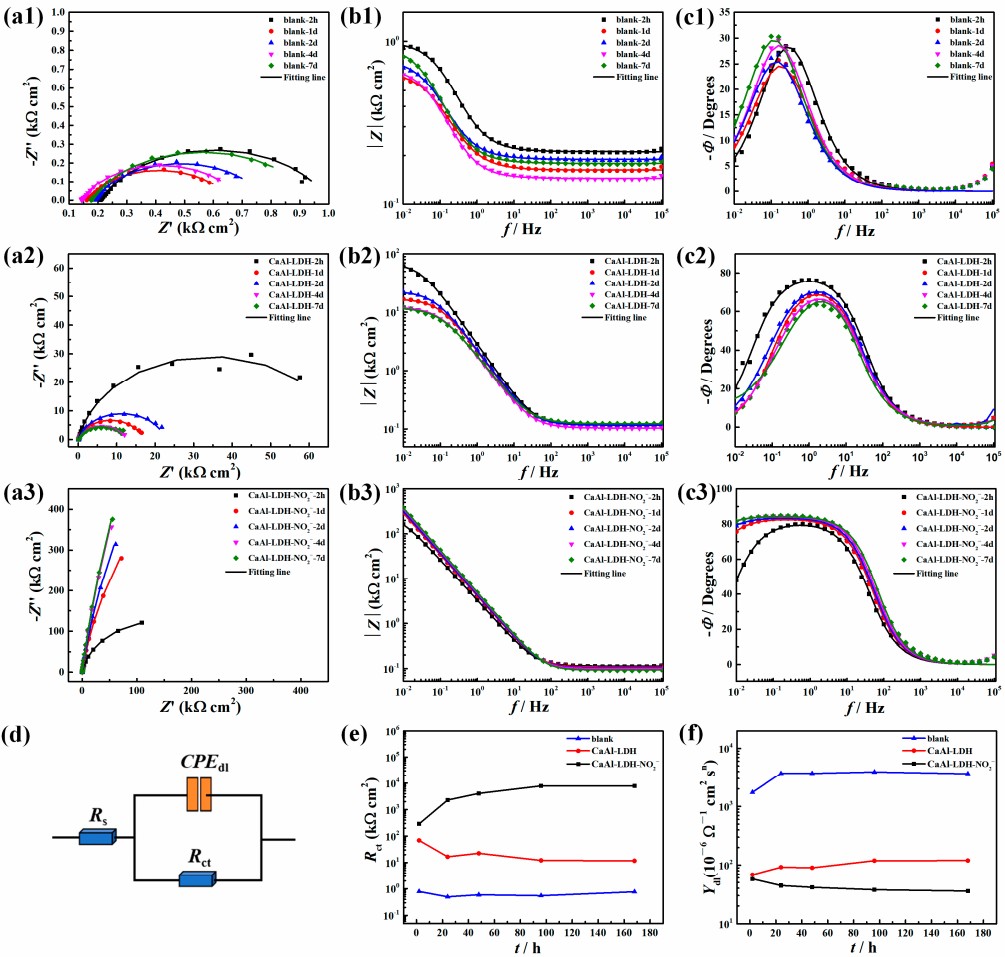

**Figure 6.** EIS results after immersed in 0.02 mol/L NaCl without (**a1–c1**) and with addition of 5 g/L CaAl-LDH (**a2–c2**) and CaAl-LDH-NO$_2$$^-$ (**a3–c3**); Equivalent circuit used to fit the EIS data (**d**); The value of $R_{ct}$ changes along with the immersion time (**e**); The value of $Y_{dl}$ changes with the immersion time (**f**).

The fitting results of the EIS data in Figure 6 are shown in Table 3. All the related electrochemical parameters are presented. According to Table 3, $Y_{dl}$ decreased and $n_{dl}$ increased remarkably in the case of LDH, in comparison with the blank sample, while this trend continued further with n values higher than 0.9 when LDH-NO$_2$$^-$ was incorporated, indicating that a compact and denser surface can be formed. In addition, the $R_{ct}$ value of the blank sample remained $7.69 \times 10^2$ Ω cm$^2$ after immersion for 7 d, while this value of

the sample with 2% LDH was $1.17 \times 10^4$ $\Omega$ cm$^2$. It is worth noting that the $R_{ct}$ value of the sample with 2% LDH-NO$_2^-$ was $7.77 \times 10^6$ $\Omega$ cm$^2$ after 7 d. The dramatic increase in the $R_{ct}$ value demonstrated a superior corrosion protection effect of LDH-NO$_2^-$ towards carbon steel in 0.02 M NaCl.

**Table 3.** The fitting results of EIS in Figure 6 based on the adopted equivalent circuit.

| Samples | Immersion Period | $CPE_{dl}$ | | $R_{ct}$ ($\Omega$ cm$^2$) |
|---|---|---|---|---|
| | | $Y_{dl}$ ($10^{-6}\Omega^{-1}$ cm$^{-2}$ s$^n$) | $n_{dl}$ | |
| Blank | 2 h | $1.74 \times 10^{-3}$ | 0.76 | $7.89 \times 10^2$ |
| | 1 d | $3.70 \times 10^{-3}$ | 0.73 | $4.98 \times 10^2$ |
| | 2 d | $3.71 \times 10^{-3}$ | 0.75 | $5.96 \times 10^2$ |
| | 4 d | $3.92 \times 10^{-3}$ | 0.76 | $5.51 \times 10^2$ |
| | 7 d | $3.67 \times 10^{-3}$ | 0.75 | $7.69 \times 10^2$ |
| CaAl-LDH | 2 h | $6.82 \times 10^{-5}$ | 0.89 | $6.88 \times 10^4$ |
| | 1 d | $9.22 \times 10^{-5}$ | 0.87 | $1.66 \times 10^4$ |
| | 2 d | $9.00 \times 10^{-5}$ | 0.86 | $2.25 \times 10^4$ |
| | 4 d | $1.19 \times 10^{-4}$ | 0.84 | $1.21 \times 10^4$ |
| | 7 d | $1.20 \times 10^{-4}$ | 0.83 | $1.17 \times 10^4$ |
| CaAl-LDH-NO$_2^-$ | 2 h | $5.74 \times 10^{-5}$ | 0.91 | $2.90 \times 10^5$ |
| | 1 d | $4.44 \times 10^{-5}$ | 0.93 | $2.29 \times 10^6$ |
| | 2 d | $4.13 \times 10^{-5}$ | 0.93 | $4.06 \times 10^6$ |
| | 4 d | $3.76 \times 10^{-5}$ | 0.94 | $7.79 \times 10^6$ |
| | 7 d | $3.58 \times 10^{-5}$ | 0.94 | $7.77 \times 10^6$ |

### 3.3. The Release Curve of NO$_2^-$ and the Chloride Concentration Decreasing Curve

The anion exchange process between the intercalated NO$_2^-$ in the LDH gallery and the Cl$^-$ in the environment were investigated by obtaining the release curve of NO$_2^-$ from CaAl-LDH-NO$_2^-$ and the decreasing concentration curve of chloride anions in 0.02 mol/L NaCl solutions with 5 g/L CaAl-LDH-NO$_2^-$ (Figure 7). As shown in Figure 7a, the released NO$_2^-$ accumulated rapidly in the initial period of the first 4 h and the release rate declined gradually with the increased immerse time. After 48 h, the concentration of NO$_2^-$ reached a plateau of higher than 18 mmol/L. However, the anion exchange reaction does not reach a dynamic equilibrium until 48 h, indicating a controlled release of inhibitor of CaAl-LDH-NO$_2^-$ in a long period. Correspondingly, the concentration of detected chloride in the solution was decreasing obviously, as shown in Figure 7b. Similarly, a sharp decrease at the initial immersion period could be observed, and then the decreasing rate dropped. It can be seen that the chloride concentration still did not reach a steady platform after 48 h. It is clear that the release curve of NO$_2^-$ and the adsorption curve of chloride were in good agreement with each other, and the rapid anion-exchange reaction in the initial immersion period reflected the burst effect of the anion exchange reaction [22,31]. It is worth noting that the amount of released NO$_2^-$ was much larger than that of the adsorbed Cl$^-$. This phenomenon can probably be attributed to the fact that NO$_2^-$ was not only released from the interlayer space through the anion-exchange reaction, the adsorbed NO$_2^-$ on the LDH surface also went into the solution under rigorous stirring.

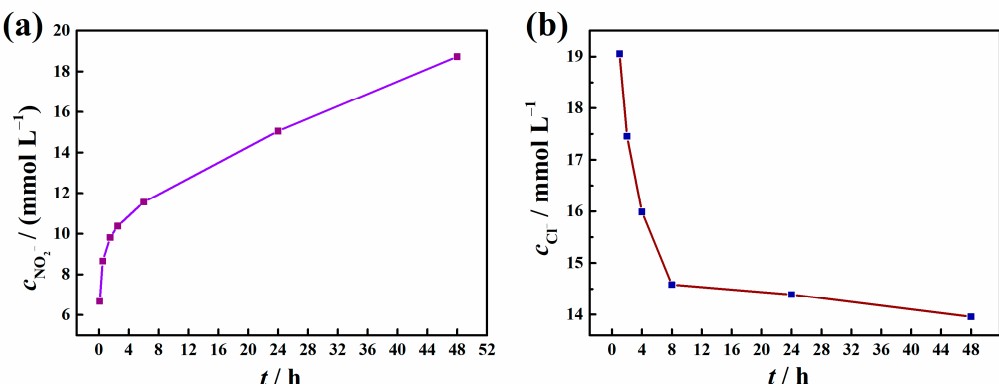

**Figure 7.** (**a**) The release curve of $NO_2^-$ from CaAl-LDH-$NO_2^-$ and (**b**) The decreasing concentration curve of chloride anions in 0.02 mol/L NaCl solutions with 5 g/L CaAl-LDH-$NO_2^-$.

*3.4. Corrosion Protection Ability of Epoxy Coatings with the Synthesized LDH Powders in 3.5 wt.% NaCl Solution*

After applying the epoxy coating on the carbon steel plate, the EIS spectra were measured during long immersion of 14 d. It was clear that the Nyquist plots of the blank epoxy sample presented two arcs, and two phase angle peaks can be observed on the Bode plots accordingly. This result indicated that serious corrosion happened during immersion in 3.5 wt.% NaCl solution. The equivalent circuit with two time constants in Figure 8j was used to simulate the EIS data. One time constant corresponded to the epoxy coatings and the other one was related to the electrochemical charge transfer reaction. $R_c$ is the coating resistance, and $CPE_c$ is the coating capacitance. $R_{ct}$ is the charge transfer resistance, and $CPE_c$ is the double layer capacitance. As for the epoxy coating with addition of 2% CaAl-LDH, the shape of the EIS spectra changed obviously. It can be seen that only one arc appeared in the Nyquist diagram and only one wide phase angle peak could be found on the Bode diagram. The other time constant did not appear, which can probably be due to the fact that corrosion did not occur due to the protection effect of the incorporated CaAl-LDH. The equivalent with one time constant was used to fit the EIS data and this time constant can be ascribed to the epoxy coatings. According to the figures in Figure 8g–i, one time constant can also be observed, and corrosion did not occur in this system. As a result, the equivalent circuit in Figure 8k can be used to simulate the EIS data. Based on the fitted data, the changes of $R_c$ values of different systems with the immersion time were listed in Figure 8l. The $R_c$ value of epoxy sample with CaAl-LDH was two orders of magnitude higher than the blank sample, while the $R_c$ value of epoxy sample with CaAl-LDH-$NO_2^-$ was one order of magnitude higher than the sample with CaAl-LDH, further suggesting the effective corrosion protection of CaAl-LDH-$NO_2^-$. In addition to the physical barrier effect of the incorporated LDH plates caused by the increased tortuosity, the presence of the nitrites in this system was able to prevent corrosion initiation and propagation effectively. The fitting results of EIS in Figure 8 based on the adopted equivalent circuit are shown in Table 4. Although the values of $Y_c$ and $n_c$ of the coated sample with 2% CaAl-LDH-$NO_2^-$ were similar to that with 2% CaAl-LDH during the immersion period, the $R_c$ values showed distinguished difference. After immersion for 14 d, the $R_c$ value of the sample with 2% CaAl-LDH was $1.05 \times 10^8$ k$\Omega$ cm$^2$, while this value was found to be $1.01 \times 10^9$ k$\Omega$ cm$^2$ for CaAl-LDH-$NO_2^-$. This remarkable increase in coating resistance can be mainly ascribed to the strong passivation effect of nitrites.

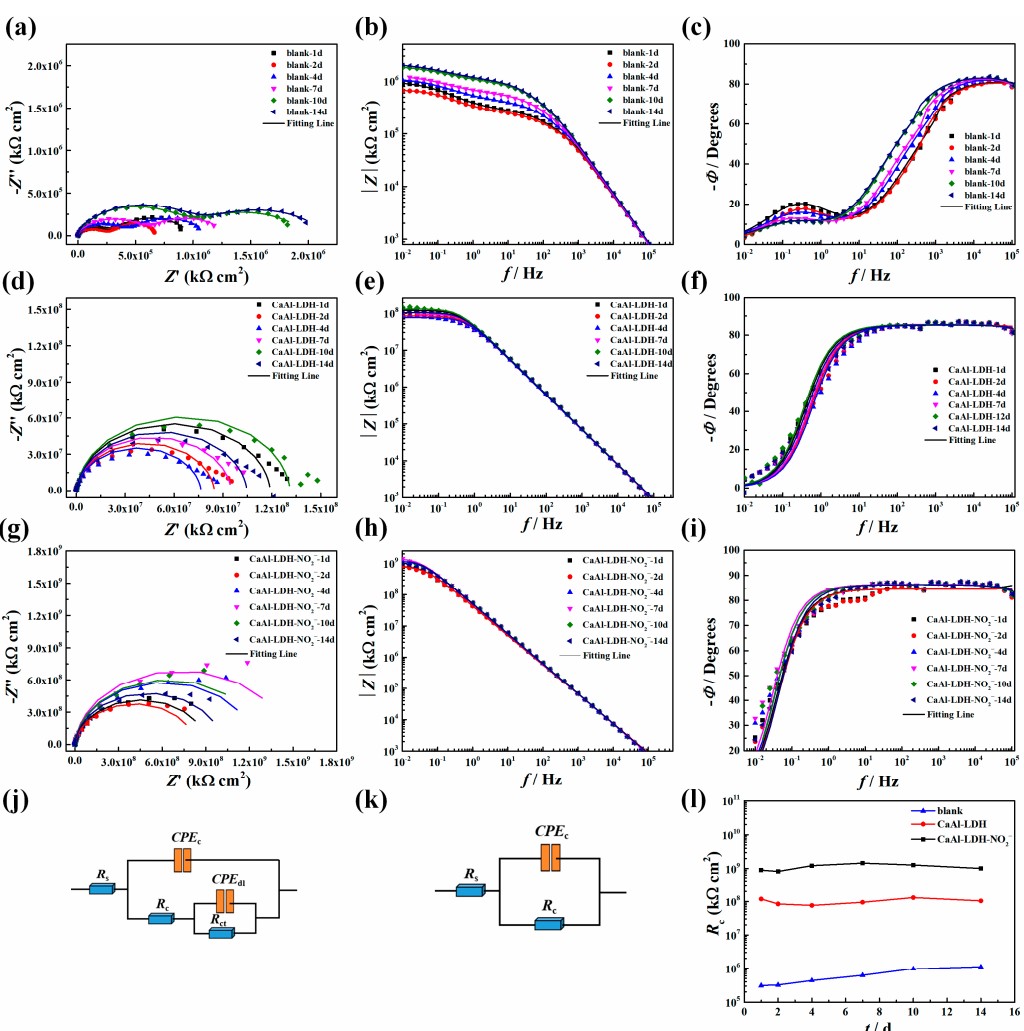

**Figure 8.** (**a–c**) The Nyquist and Bode plots of blank epoxy coatings; (**d–f**) Epoxy coatings with addition of 2% CaAl-LDH; (**g–i**) Epoxy coatings with addition of 2% CaAl-LDH-NO$_2^-$; (**j**) Equivalent circuits used to fit EIS data of blank epoxy coatings; (**k**) Equivalent circuits used to fit EIS data of epoxy coatings with 2% CaAl-LDH and 2% CaAl-LDH-NO$_2^-$; (**l**) $R_c$ changes along with immersed time.

**Table 4.** The fitting results of EIS in Figure 8 based on the adopted equivalent circuit.

| Samples | Immersion Period | $CPE_c$ | | $R_c$ (kΩ cm$^2$) | $CPE_{dl}$ | | $R_{ct}$ |
|---|---|---|---|---|---|---|---|
| | (d) | $Y_c$ (10$^{-6}$ Ω$^{-1}$ cm$^{-2}$ s$^n$) | $n_c$ | | $Y_{dl}$ (10$^{-6}$ Ω$^{-1}$ cm$^{-2}$ s$^n$) | $n_{dl}$ | (kΩ cm$^2$) |
| Blank | 1 | $4.05 \times 10^{-5}$ | 0.68 | $3.09 \times 10^5$ | $1.80 \times 10^{-3}$ | 0.66 | $6.99 \times 10^5$ |
| | 2 | $2.16 \times 10^{-5}$ | 0.59 | $3.23 \times 10^5$ | $2.40 \times 10^{-3}$ | 0.73 | $4.14 \times 10^5$ |
| | 4 | $6.51 \times 10^{-5}$ | 0.72 | $4.36 \times 10^5$ | $1.70 \times 10^{-3}$ | 0.61 | $7.32 \times 10^5$ |
| | 7 | $2.78 \times 10^{-5}$ | 0.70 | $6.20 \times 10^5$ | $1.96 \times 10^{-3}$ | 0.62 | $7.14 \times 10^5$ |
| | 10 | $2.76 \times 10^{-5}$ | 0.76 | $9.70 \times 10^5$ | $1.21 \times 10^{-3}$ | 0.56 | $1.06 \times 10^6$ |
| | 14 | $1.71 \times 10^{-5}$ | 0.72 | $1.11 \times 10^6$ | $1.30 \times 10^{-3}$ | 0.59 | $1.09 \times 10^6$ |
| CaAl-LDH | 1 | $3.45 \times 10^{-6}$ | 0.95 | $1.91 \times 10^8$ | - | - | - |
| | 2 | $3.62 \times 10^{-6}$ | 0.95 | $8.48 \times 10^7$ | - | - | - |
| | 4 | $3.69 \times 10^{-6}$ | 0.95 | $7.65 \times 10^7$ | - | - | - |
| | 7 | $3.46 \times 10^{-6}$ | 0.95 | $9.51 \times 10^7$ | - | - | - |
| | 10 | $3.42 \times 10^{-6}$ | 0.95 | $1.31 \times 10^8$ | - | - | - |
| | 14 | $3.53 \times 10^{-6}$ | 0.95 | $1.05 \times 10^8$ | - | - | - |

**Table 4.** *Cont.*

| Samples | Immersion Period | $CPE_c$ | | $R_c$ (kΩ cm²) | $CPE_{dl}$ | | $R_{ct}$ |
|---------|------------------|---------|--|----------------|------------|--|----------|
| CaAl-LDH-NO$_2^-$ | 1 | $4.07 \times 10^{-6}$ | 0.94 | $9.08 \times 10^8$ | - | - | - |
| | 2 | $4.09 \times 10^{-6}$ | 0.94 | $8.28 \times 10^8$ | - | - | - |
| | 4 | $3.40 \times 10^{-6}$ | 0.96 | $1.23 \times 10^9$ | - | - | - |
| | 7 | $3.33 \times 10^{-6}$ | 0.96 | $1.46 \times 10^9$ | - | - | - |
| | 10 | $3.32 \times 10^{-6}$ | 0.96 | $1.28 \times 10^9$ | - | - | - |
| | 14 | $3.33 \times 10^{-6}$ | 0.96 | $1.01 \times 10^9$ | - | | - |

### 3.5. Coating Morphology Characterization

The top-view and side-view of the epoxy coating with and without the addition of CaAl-LDH and CaAl-LDH-NO$_2^-$ are shown in Figure 9. According to the top-view morphology in Figure 9(a1–c1), almost no obvious aggregates can be found in the blank epoxy coating (Figure 9(a1)) and the epoxy coating with 2% CaAl-LDH-NO$_2^-$ (Figure 9(c1)), while some aggregates could be found on the surface of the epoxy coating with 2% CaAl-LDH (Figure 9(b1)). This result demonstrated that the modification of NO$_2^-$ probably enhanced the compatibility of CaAl-LDH with the coating to some extent. The side-view morphology in Figure 9(a2–c2) showed richer information. It can be seen from Figure 9(a2) that some holes appeared in the blank epoxy coating, while the addition of CaAl-LDH filled the holes to some extent, and the compactness of the coating was increased. The addition of CaAl-LDH-NO$_2^-$ further enhanced the compactness of the coating and no holes or aggregates could be observed in the side-view morphology.

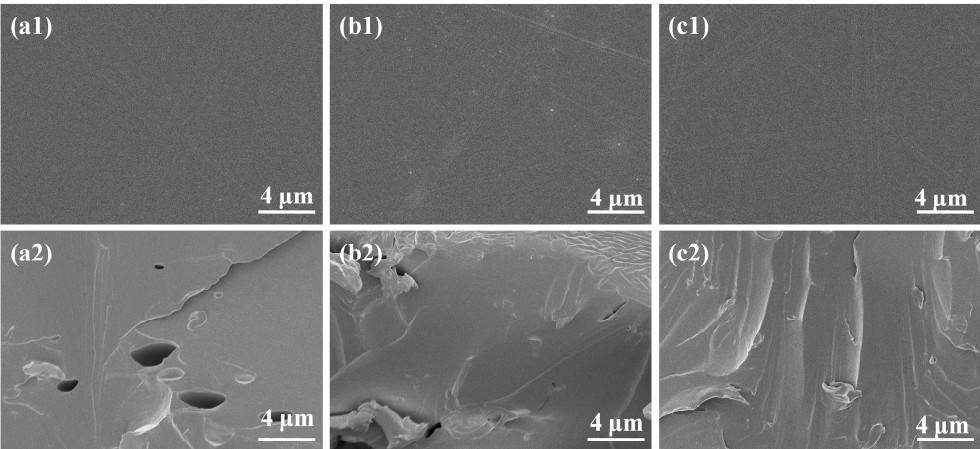

**Figure 9.** The top-view (**a1–c1**) and side-view (**a2–c2**) morphology of blank epoxy coating (**a1,a2**); epoxy coating with 2% CaAl-LDH (**b1,b2**); epoxy coating with 2% CaAl-LDH-NO$_2^-$ (**c1,c2**).

The 3D morphology and roughness of the coating surface were characterized by laser microscopy, which are shown in Figure 10. It was obvious that the surface of the epoxy coating with 2% CaAl-LDH was rougher than that of the blank epoxy coating and the coating with 2% CaAl-LDH-NO$_2^-$. The changes of the scale bar were in good accordance with the surface morphology, and a larger scale bar usually corresponded to a rougher surface. According to the obtained roughness results, the $R_a$ value of the blank epoxy coating, the epoxy coating with 2% CaAl-LDH and the epoxy coating with 2% CaAl-LDH-NO$_2^-$ was measured to be 0.309 ± 0.02 μm, 0.409 ± 0.08 μm and 0.316 ± 0.02 μm, respectively. This result was in good agreement with the SEM morphology result in Figure 9. It demonstrated that the addition of CaAl-LDH has a negative influence on the compactness of the epoxy coating, and the modification of CaAl-LDH using NO$_2^-$ could improve its compatibility with the epoxy coating; however, the related reasons remain to be further investigated.

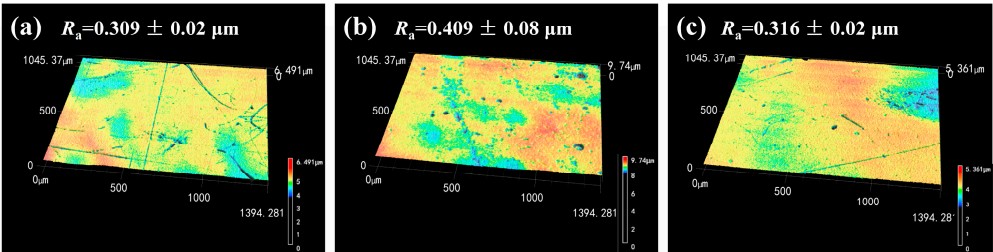

**Figure 10.** The 3D morphology and roughness of (**a**) Blank epoxy coating; (**b**) Epoxy coating with 2% CaAl-LDH; (**c**) Epoxy coating with 2% CaAl-LDH-$NO_2^-$.

*3.6. The Morphology and Corrosion Product Characterization of the Scratched Coating Samples after Salt Spray Test*

In order to further study the corrosion protection capability of the added LDH-$NO_2^-$, the coated samples were scratched artificially and subjected to salt spray test for 7 d. The rust was removed and the optical images were captured, and are shown in Figure 11(a1–c1,a2–c2). It can be seen that the corrosion was quite serious on the scratched area of the blank epoxy coating. For the coating with 2% CaAl-LDH, corrosion was less severe, probably due to the chloride trapping effect based on the anion-exchange reaction between the intercalated anions and the chlorides, and also the physical barrier effect, as evidenced in Figure 9(b2). For the coating with 2% CaAl-LDH-$NO_2^-$, corrosion was further inhibited due to the effective inhibiting effect of the released $NO_2^-$ and the enhanced physical barrier effect. According to the EDS results in Figure 11, the content of the Fe element in the scratched area of the epoxy coating with 2% CaAl-LDH increased compared to the blank epoxy coating, where the atomic percentage of Fe increased from 26.6% to 41.36%. It is worth noting that this value further increased to 47.77% in the case of the epoxy coating with 2% CaAl-LDH-$NO_2^-$.

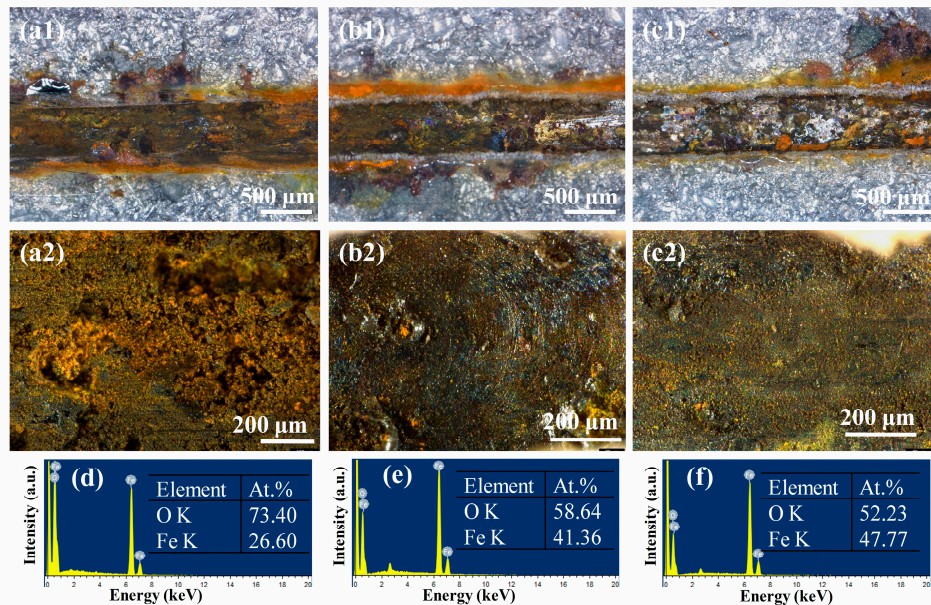

**Figure 11.** The optical images the scratched area of the coating sample after salt spray test for 7 days (**a1**,**a2**) blank epoxy coatings; (**b1**,**b2**) epoxy coatings with 2% CaAl-LDH; (**c1**,**c2**) epoxy coatings with 2% CaAl-LDH-$NO_2^-$. (**d**) The EDS results of the corrosion product of scratched area of the blank epoxy coatings; (**e**) The EDS results of the corrosion product of the scratched area of the epoxy coatings with 2% CaAl-LDH; (**f**) The EDS results of the corrosion product of the scratched area of the epoxy coatings with 2% CaAl-LDH-$NO_2^-$.

In order to further analyze the composition of the rust of different scratched samples, Raman spectra were recorded, which is shown in Figure 12. For the blank epoxy coating, the peaks around 400 and 1300 cm$^{-1}$ belonged to $\alpha$-FeOOH and $\gamma$-FeOOH, respectively. The peak at 285 cm$^{-1}$ was attributed to $Fe_2O_3$. As for the epoxy coating with addition of 2% CaAl-LDH and CaAl-LDH-NO$_2$$^-$, the peak around 400 cm$^{-1}$ disappeared completely, and the peak at 1300 cm$^{-1}$ decreased notably, while the peak around 650–680 cm$^{-1}$ appeared obviously, which can be ascribed to $Fe_3O_4$ [39,40]. The intensity of this peak further increased for the epoxy coating with 2% CaAl-LDH-NO$_2$$^-$ in comparison with that with 2% CaAl-LDH, while the peak corresponding to r-FeOOH further weakened. The changes of the main corrosion product from a-FeOOH and r-FeOOH to $Fe_3O_4$ demonstrated that the existence of CaAl-LDH and CaAl-LDH-NO$_2$$^-$ was able to raise the corrosion protection ability of the coating. The CaAl-LDH-NO$_2$$^-$ was able to provide more effective protection due to the superior inhibiting effect of NO$_2$$^-$.

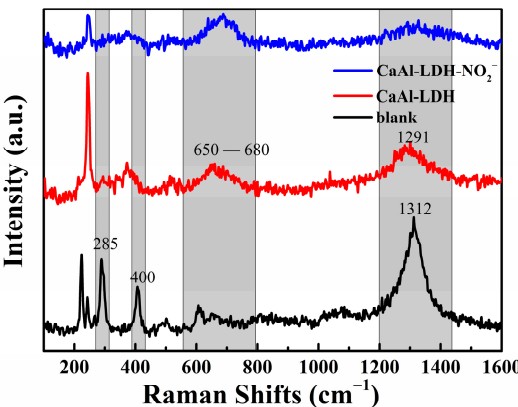

**Figure 12.** Raman spectra of the rust obtained from the sample in Figure 11.

### 3.7. Self-Healing Performance of the Coatings

To investigate the corrosion protection mechanism of the synthesized LDH, an artificial hole was made by a sharp metal tip on the coating surface. The spectra at 1, 24, and 48 h were recorded and shown in Figure 13. It can be seen clearly that all the coatings demonstrated a low corrosion resistance in the hole and a higher corrosion resistance in the surrounding area. It is worth to note that obvious differences can be detected between various coatings. For the blank epoxy coating, serious corrosion occurred at the initial period of 1 h and no remarkable changes can be found until the immersion of 48 h. When CaAl-LDH was included in the epoxy coating, severe corrosion also occurred in this system due to the contact between the exposed metal substrate at the artificial hole and the aggressive NaCl solution, while corrosion was less serious than that of the blank epoxy coating, probably due to the chloride adsorption effect of LDH. As for the coating with CaAl-LDH-NO$_2$$^-$, almost no corrosion could be found at the immersion of 1 h; this attractive phenomenon can be attributed to the healing effect of the release of nitrites from CaAl-LDH-NO$_2$$^-$. Passive film was formed on the metal substrate surface as nitrites could act a strong oxidizer, and the corrosion site was healed. However, after immersion for 4 h, corrosion happened again in this healed area, probably due to the fact that the passive film was not dense enough, and it was broken at the strong attack of chlorides. Corrosion further propagated after immersion for 48 h. This result inspired us to state that the content of CaAl-LDH-NO$_2$$^-$ should be further optimized to obtain a more effective protection film at the corrosion area, or other more effective corrosion inhibitors could be loaded in LDH to achieve such a purpose in future work.

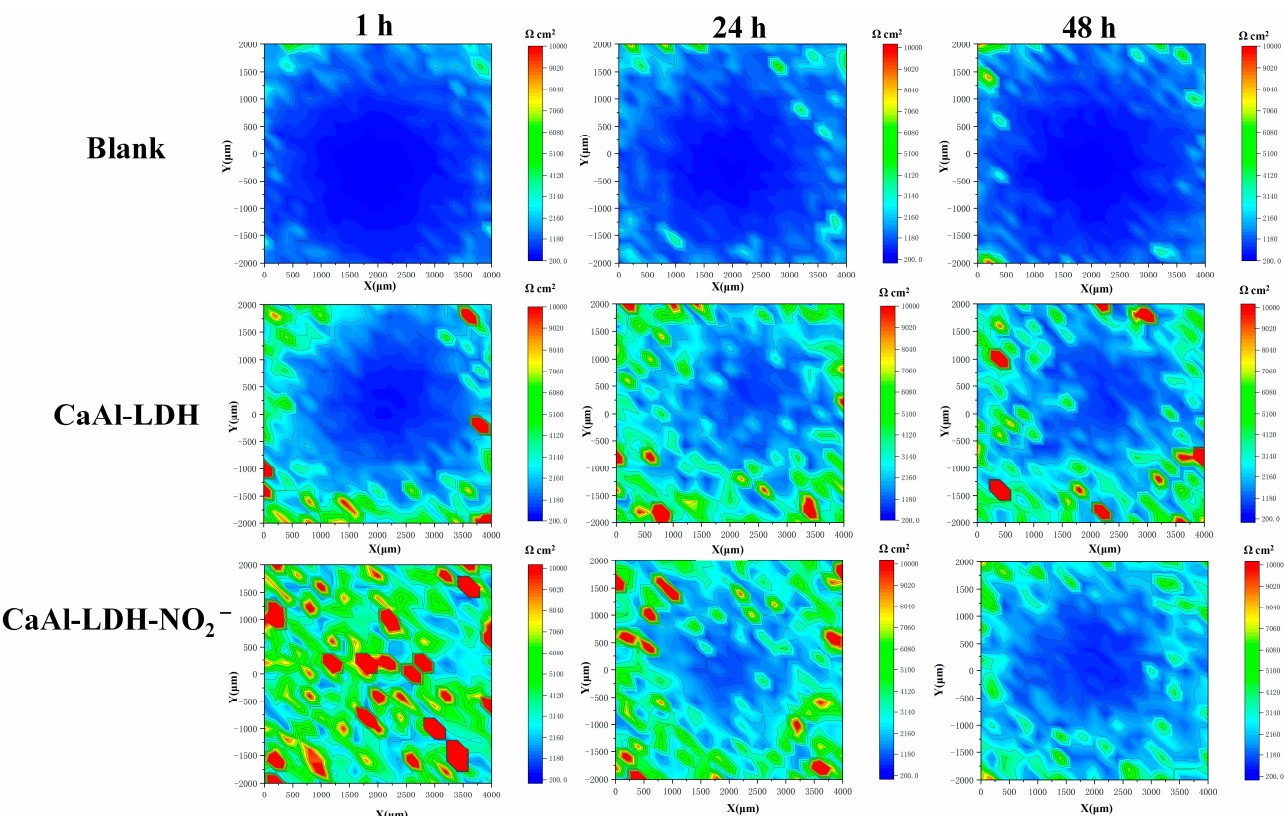

**Figure 13.** The LEIS maps of blank epoxy coatings, epoxy coatings with 2% CaAl-LDH, epoxy coatings with 2% CaAl-LDH-NO$_2^-$ during an immersion period of 1, 24, and 48 h in 3.5 wt.% NaCl.

*3.8. Corrosion Protection Mechanism*

The corrosion protection mechanism of CaAl-LDH-NO$_2^-$ is shown in Figure 14. As presented in this figure, water molecules and chlorides can be regarded as two main factors resulting in the initiation of corrosion. The enhanced corrosion protection ability of the epoxy coating with the addition of CaAl-LDH-NO$_2^-$ can be attributed to the following two points. On one hand, the existence of LDH platelets was able to improve the tortuosity of the transportation path of water molecules and chlorides [41,42]; therefore, the time when the aggressive ions and water molecules reached the underlying substrate would be delayed to some extent. On the other hand, the anion-exchange reaction occurred between the intercalated nitrites in the LDH interlayer space and the aggressive chlorides in the external environment. As a result, the chlorides were adsorbed and the inhibitor nitrites were released [18,43]. The chloride concentration was finally decreased and the attack towards the substrate was mitigated. Furthermore, the released nitrites act as inhibitors and heal the corrosion sites, and corrosion propagation can be prevented in this way. The protective film was formed mainly according to the following equation in neutral or basic solutions [44,45]:

$$2Fe^{2+} + 2OH^- + 2NO_2^- \rightarrow 2NO + \gamma\text{-}Fe_2O_3 + H_2O \qquad (2)$$

Based on the EIS results in Figure 8, it can be concluded that the anion exchange reaction is able to play a significant role in the overall enhanced corrosion protection effect in addition to the physical barrier effect of LDH platelets, which differs from the case where no inhibitors are intercalated [28]. The inhibitor type, loading content of inhibitor, and the additive amount of LDH powders can be further optimized to obtain a better corrosion protection effect. In addition, the formation of CaMoO$_4$ during molybdate modification demonstrated that the successful synthesis of active LDH inhibitor container was determined by two factors, including the metal ion type of LDH and the inhibitor type.

If chemical reaction occurs between the metal cations and the inhibitor anions, then the inhibitor anions probably cannot play a significant role in corrosion inhibition.

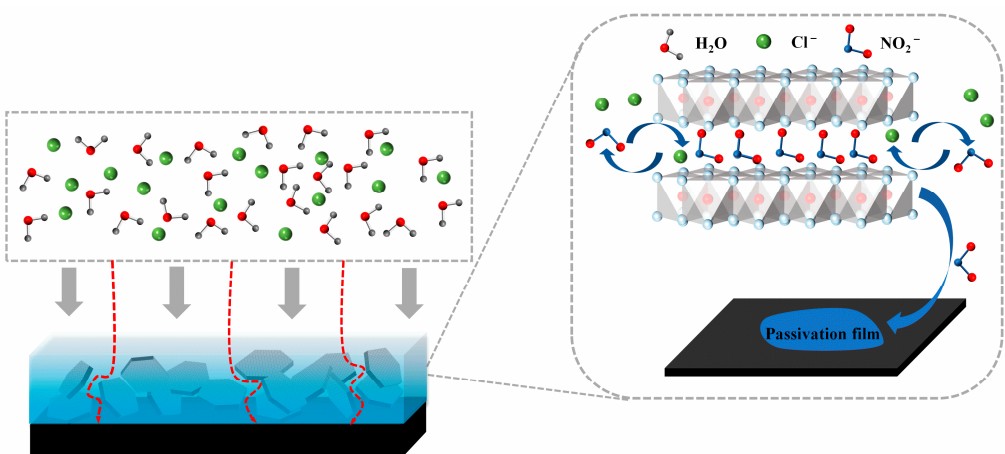

**Figure 14.** Schematic presentation of the corrosion protection mechanism of CaAl-LDH-$NO_2^-$.

## 4. Conclusions

In this work, CaAl-LDH modified with nitrites and molybdates was synthesized and the SEM result indicated plate-like morphology, and the XRD result verified the successful intercalation of nitrites in the LDH gallery, while LDH after molybdate modification underwent structure destruction, and $CaMoO_4$ was finally formed. The corrosion protection ability of CaAl-LDH-$NO_2^-$ in 0.02 M NaCl solution and in the epoxy coating was evaluated by EIS, and the results demonstrated that the addition of CaAl-LDH-$NO_2^-$ with a concentration of 5 g/L was able to provide efficient corrosion protection for carbon steel, while LDH modified by molybdates almost could not present a corrosion inhibition effect. Furthermore, 2% (weight vs. epoxy resin) incorporation in the epoxy coating was able to enable the coating to present enhanced coating corrosion resistance with one order of magnitude higher than the coating with 2% CaAl-LDH without inhibitors. The corrosion protection effect was ascribed to the nitrite release and the chloride trapping based on the anion exchange micro-reaction and the improved physical barrier effect in the coating. It should be noted that subsequent work will be carried out to optimize inhibitor type, loading content of inhibitor, and the additive amount of LDH powders to obtain better corrosion protection in the near future. In addition, the failure of LDH modification using molybdate in this work can also give researchers some inspiration in designing active LDH microcontainers intercalated with inhibitors.

**Supplementary Materials:** The following supporting information can be downloaded at: https://www.mdpi.com/article/10.3390/coatings13071166/s1, Figure S1: The EIS results of carbon steel in 0.02 M NaCl solution with 5 g/L $MoO_4^{2-}$.

**Author Contributions:** Conceptualization, Y.C.; Software, X.Z.; Validation, L.Z.; Investigation, J.X., X.Z. and L.Z.; Data curation, J.X.; Writing—original draft, Y.C.; Writing—review & editing, J.W., Y.C. and C.H.; Supervision, J.W., Y.C. and C.H.; Project administration, K.C. All authors have read and agreed to the published version of the manuscript.

**Funding:** This work was funded by the Xiamen Natural Science Foundation Project (Grant No. 3502Z20227160) and the State Key Laboratory for Marine Corrosion and Protection Foundation (Grant No. JS220902).

**Institutional Review Board Statement:** Not applicable.

**Informed Consent Statement:** Not applicable.

**Data Availability Statement:** Not applicable.

**Conflicts of Interest:** The authors declare no conflict of interest.

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
