# Peer review of "Corrosion Protection Mechanism Study of Nitrite-Modified CaAl-LDH in Epoxy Coatings"

_coatings, doi:10.3390/coatings13071166_

Round 1
Reviewer 1 Report
Ms. Ref. No.: coatings-2463509
Title: "Corrosion protection mechanism study of nitrite modified CaAl-LDH in epoxy coatings".
This work presents the improving the corrosion performance of Q235 carbon steel by novel CaAl-LDH in epoxy coatings. The results reported in this paper could be valuable for publication. However, several points have to be reconsidered for “Major” revision. In my opinion, the following points have to be considered:
- The experimental section is poorly written. The purity and manufacturer of reagents should be given, especially for the electrolyte reagents. Many essential details of surface characterization and evaluation are missing. The description is virtually reduced to listing the methods and instruments used, rather than details of procedures and subsequent analysis. Many details in electrochemical studies (immersion time, applied potential, applied frequencies, etc.) have not been mentioned.
- In Figures 2 and 3, the x- and y-axes of EDS spectra should be labeled with related names and units.
- How the authors recognized the different phases in the analysis of XRD in Figure 4? It is suggested to mention whether the phases are matched with their respective JCPDS cards or any other method.
- Why were 7 days chosen for the salt spray test? What standard was used for the spraying test? Can the coated steel samples still have good performance after experiencing the salt spray test a longer time?
- The proposed corrosion protection mechanism in a scientific experimental paper should be based on references.
- Typos and minor grammar errors are common in the manuscript. A considerable number of sentences are not comprehensible, some of which can be due to typos or grammar errors. About the manuscript, it is recommended to recheck the grammatical errors or the subscript errors. Authors should pay more attention to singular/plural nouns. Also, they should control the spell check/ punctuation of words and sentences. In addition, spaces should be added between words and numbers. Please fix the typographical and eventual language problems in the paper.
- The technical terms appear in different forms and are sometimes misspelled. Abbreviations are not always defined at first occurrence. The formatting of the text is inconsistent, no attention was paid to the correct use of indices or letter formats (e.g. the formatting of references must meet the expectations of the journal: https://www.mdpi.com/authors/references).
The English of the whole paper is good, but some errors could still be found. Therefore, the English of the paper should be reviewed.
Author Response
- The experimental section is poorly written. The purity and manufacturer of reagents should be given, especially for the electrolyte reagents. Many essential details of surface characterization and evaluation are missing. The description is virtually reduced to listing the methods and instruments used, rather than details of procedures and subsequent analysis. Many details in electrochemical studies (immersion time, applied potential, applied frequencies, etc.) have not been mentioned.
Response to comment: Thank you for your comment. We have improved the experimental section in the revised version. The purity of the reagents was added in the revised manuscript and the manufacturer information was included in the original version. The details of EIS and LEIS including immersion time, applied potential and applied frequencies were already contained in the original version, and more detailed description about the surface characterization and evaluation was added.
- In Figures 2 and 3, the x- and y-axes of EDS spectra should be labeled with related names and units.
Response to comment: Thank you for your advice. We have added the name and units for the x- and y-axes of EDS spectra in Fig.1 and 3 in the revised version.
- How the authors recognized the different phases in the analysis of XRD in Figure 4? It is suggested to mention whether the phases are matched with their respective JCPDS cards or any other method.
Response to comment: Thank you for your comment. For the identification of CaMoO4, the PDF number of 29-0351 was provided. In addition, we have added the PDF number of for the recognition of CaCO3 in the revised version (PDF47-1743). As for the identification of CaAl-LDH, the related reference was included in the manuscript (Ref [29]).
- Why were 7 days chosen for the salt spray test? What standard was used for the spraying test? Can the coated steel samples still have good performance after experiencing the salt spray test a longer time?
Response to comment: Thank you for your comment. Indeed when we are performing the salt spray test, we wanted the coated steel samples to experience a longer time of the salt spray test. However, corrosion with different degrees was observed on the scratched area of these three different kinds of coated samples after 7 days, even on the sample with LDH-NO2-. This phenomenon can be possibly attributed to the large size of the scratched area and the limited release of nitrite inhibitor caused by the inadequate addition of LDH-NO2- in the coating. In our future work, we will focus on the optimization of the addition of LDH-NO2- in the epoxy coatings to obtain better corrosion inhibition effect.
- The proposed corrosion protection mechanism in a scientific experimental paper should be based on references.
Response to comment: Thank you for your comment. Related literatures have been added in this part.
- Typos and minor grammar errors are common in the manuscript. A considerable number of sentences are not comprehensible, some of which can be due to typos or grammar errors. About the manuscript, it is recommended to recheck the grammatical errors or the subscript errors. Authors should pay more attention to singular/plural nouns. Also, they should control the spell check/ punctuation of words and sentences. In addition, spaces should be added between words and numbers. Please fix the typographical and eventual language problems in the paper.
Response to comment: Thank you for your suggestions. We have made a careful check towards the whole manuscript. All the errors were corrected now.
- The technical terms appear in different forms and are sometimes misspelled. Abbreviations are not always defined at first occurrence. The formatting of the text is inconsistent, no attention was paid to the correct use of indices or letter formats (e.g. the formatting of references must meet the expectations of the journal: https://www.mdpi.com/authors/references).
Response to comment: Thank you for your comment. The technical terms were unified and the abbreviations were defined at the first presence now. In addition, the format of the cited references was revised according to the requirement of the journal in the revised manuscript.
Comments on the Quality of English Language
The English of the whole paper is good, but some errors could still be found. Therefore, the English of the paper should be reviewed.
Response to comment: Thank you for your suggestions. The English of the paper was reviewed carefully. The errors were corrected now.

Reviewer 2 Report
1. Characterization of the synthesized LDH powders. Figure 2. In this image the microns do not appear.
2. Figure 3. The scale of X axis must be amplified from 0-12 keV to try to detect the presence of nitrogen.
3. Conclusions. In this section, it would be necessary to point out that there will be other subsequent works to try to optimize the content of LDH-NO2- and thus obtain better corrosion protection.
English is quite correct
Author Response
Comments from reviewer 2
- Characterization of the synthesized LDH powders. Figure 2. In this image the microns do not appear.
Response to comment: Thank you for your comment. The scale bar in Fig.2 has been added.
- Figure 3. The scale of X axis must be amplified from 0-12 keV to try to detect the presence of nitrogen.
Response to comment: Thank you for your advice. The scale of X axis has been amplified from 0-12 keV. Nitrogen was detected and related analysis was added in the revised manuscript.
- Conclusions. In this section, it would be necessary to point out that there will be other subsequent works to try to optimize the content of LDH-NO2- and thus obtain better corrosion protection.
Response to comment: Thank you for your advice. The following sentence was added in the part of Conclusion: “It should be noted that subsequent work will be carried out to optimize inhibitor type, loading content of inhibitor and the additive amount of LDH powders to obtain better corrosion protection in the near future.”

Reviewer 3 Report
Please find the attached file

Minor editing required
Author Response
Comments from Reviewer 3
The authors used an LDH-inhibitor system to improve anti-corrosion properties of epoxy coatings. We think it can be published in the Journal of Coatings after a major revision. Some advice is given as follows:
1- The first line of the abstract needs to be revised.
Response to comment: Thank you for your comment. The first line of the abstract was revised as “In this work, nitrite intercalated CaAl-LDH was firstly prepared and its corrosion protection mechanism in epoxy coatings was investigated.”
2- The method of measuring chloride and nitrite ions in the abstract should be mentioned.
Response to comment: Thank you for your advice. The nitrite release curve and the chloride concentration change curve were obtained by UV-Vis spectroscopy and home-made Ag/AgCl probe, respectively. Related information was added in the revised version.
3- The concentration of the powder added to the epoxy coating needs to be clearly stated in the abstract (2% weight or volume?).
Response to comment: Thank you for your comment. The powder was added in the epoxy coating with a percentage of 2% (weight vs. epoxy resin). Such information was added in the revised version.
4- The quantitative results of the tests should be included in the abstract.
Response to comment: Thank you for your comment. The following quantitative results of the tests were added in the abstract.(include concrete values?)
“The fitted coating resistance of the sample with 2% LDH intercalated with nitrites was one order of magnitude higher than that with 2% LDH, and the latter one was two orders of magnitude were higher than the blank sample.”
5- For all of the mentioned 2D carriers, appropriate references should be provided in lines 36 and 37.
Response to comment: Thank you for your comment. Appropriate references have been added for all of the mentioned 2D carriers.
6- For all of the carriers, after being mentioned in the text, appropriate references should be provided in lines 45 to 50.
Response to comment: Thank you for your comment. Appropriate references have been provided accordingly.
7- The specifications of the used epoxy and hardener components need to be mentioned in the materials and methods section. The information provided is not sufficient.
Response to comment: Thank you for your suggestion. The specifications of the epoxy resin and the hardener have been added in the Experimental section.
Table 1 The main physical and chemical properties of 6101 epoxy resin
Items |
6101 epoxy resin |
Epoxy equivalent (g/eq) |
210-240 |
softening point (℃) |
12-20 |
viscosity (25℃ mPa·s 80%Xylene solution) |
200-400 |
Saponifiable chlorine (%) |
<=0.3 |
Inorganic chlorine (ppm) |
<=180 |
Table 2 The main physical and chemical properties of 2519 hardener
Items |
2519 hardener |
Viscosity (25℃ mPa·s) |
185 |
Density (25℃ g/ml) |
1.01 |
Flash point(℃) |
>100 |
Amine value (mg KOH / g) |
315 |
8- The authors have used molybdate as a corrosion inhibitor in addition to nitrate, but it is not mentioned in the title or abstract.
Response to comment: Thank you for your comment. Yes, we have tried the modification of molybdate for CaAl-LDH, however, the result indicated that it did not work. According to the SEM result in Fig.3 and the XRD result in Fig.4, molybdates can react with CaAl-LDH and CaMoO4 can be finally formed during the modification process. Therefore, most of the molybdates cannot be released in the system and therefore cannot play an active role in corrosion inhibition. This means although molybdates belong to traditional inhibitors, it may be useless in certain cases. Based on the reviewer’s comment, we have added the information related to molybdate modification in the abstract and also other sections in the revised version. However, the title kept unchanged, since the modification of LDH using molybdate failed and thus we did not add this sample into epoxy coatings. Therefore, it should be noted that the corrosion protection mechanism study was only pointed at the nitrite modified LDH. That is why the title was unchanged.
9- Lines 131, 398, and 401: cm-1 is incorrect and needs to be corrected.
Response to comment: Thank you for your comment. The mentioned mistakes have been corrected.
10- The device used for LEIS analysis should be mentioned in the materials and methods section.
Response to comment: Thank you for your comment. The device information for LEIS was added in the revised version.
11- Units of the values mentioned in lines 209 and 210 need to be stated.
Response to comment: Thank you for your comment. Thank you for your comment, The units of the values were added.
12- Line 235: there is a grammatical mistake that needs to be corrected.
Response to comment: Thank you for your comment. The grammatical mistake in this line has been corrected. This sentence has been revised as “The peak at 1352 cm-1 and 1384 cm-1 can be attributed to the carbonates and nitrates, respectively”.
13- Please explain why the peak at 1352 cm-1 has almost disappeared in the FTIR spectra of the anion-containing samples?
Response to comment: Thank you for your comment. According to your comment, we further analyzed the FTIR result in Fig.5. For CaAl-LDH-NO2-, the obvious characteristic peak of nitrites at 1270 cm-1 appeared, the small peak attributed to nitrates and carbonates at 1383 cm-1 and 1353 cm-1 can also observed. The dramatic decrease of the peak around 1353 cm-1 in comparison with that of CaAl-LDH may be probably due to the anion-exchange reaction between the carbonates in the LDH interlayer space and the nitrites in the surrounding environment during the nitrite modification process. This analysis was added in the revised manuscript.
14- Why was the EIS test started after only 2 hours? How do you know that after this time, the sample has reached a steady state? Where are the results of the OCP test?
Response to comment: Thank you for your comment. Based on our previous research experience, OCP became stable only after a short time of 0.5 h in this kind of system (Corrosion Science 126 (2017) 166–179; Journal of The Electrochemical Society, 166 (11) C3106-C3113 (2019)). Therefore, OCP result was not recorded during the experiment. According to your comment, the related sentence in the Results and discussion section has been revised as follows: The EIS results were recorded after immersion for 2 h in an immersion duration of 7 d since open circuit potential (OCP) reached a steady state only after 0.5 h.
15- The Cdl parameter was not defined, and the method of calculating of the parameters was not described.
Response to comment: Thank you for your comment. In order to better simulate the real case, CPEdl was used instead of Cdl in the revised version. The impedance of CPE can be calculated by the following expression:
ZCPE=Y-1(jw)-n
Where Y (Ω-1 cm-2 sn) is a proportional factor, n is a factor ranging from 0 to 1.
This equation was added in the revised manuscript.
16- The method of measuring ion concentrations in Figure 7 needs to be mentioned in the materials and methods section.
Response to comment: Thank you for your comment. The method of measuring ion concentrations in Figure 7 has been added in the Experimental section.
17- Line 310: replace "absorbed" with "adsorbed."
Response to comment: Thank you for your comment. "absorbed" has been replaced with "adsorbed".
18- To investigate the inhibition mechanism in the solution phase, polarization tests are more useful than EIS. Please add the results of this test to the manuscript.
Response to comment: Thank you for your comment. If we want to investigate the inhibition mechanism of an anodic, cathodic or mixed corrosion inhibitor, polarization tests would be quite useful. Rich information such as the corrosion rate, corrosion potential, Tafel slope can be obtained from the polarization test results. However, in this paper, we are aimed to study the corrosion protection mechanism of CaAl-LDH intercalated with traditional inhibitors in epoxy coatings rather than investigation of the corrosion inhibition mechanism of the inhibitors, which has been studied man years ago and was quite clear. EIS is able to provide rich information about the corrosion process of the coated samples through EIS data fitting based on the adopted equivalent circuit. Therefore, in our opinion, it is not necessary to add the polarization test results.
19- Why were different electrolyte concentrations chosen for the solution phase and coating phase sections?
Response to comment: Thank you for your comment. Yes, 0.02 mol/L NaCl solution was used for EIS measurement in solution phase while 3.5 wt.% NaCl solution was used for EIS measurement in coating phase. Two kinds of NaCl solution with different concentrations was used because the corrosion resistance against the aggressive chloride attack of different samples was different. In 0.02 mol/L NaCl solution, exposed carbon steel was used while the carbon steel coated with epoxy coatings was used in 3.5 wt.% NaCl solution. If 0.02 mol/L NaCl solution was used for the EIS measurement of the coated samples, it may take a long time to obtain the differences between the blank sample and the sample with LDH.
20- Please include the values of other electrochemical elements in the equivalent circuit in a table.
Response to comment: Thank you for your comment. The values of other electrochemical elements in the equivalent circuit were listed in Table 3 and Table 4. In addition, related analysis towards these values was also added in the revised manuscript.
21- In Figure 8 (k), Qdl should be replaced with Qc.
Response to comment: Thank you for your comment. Qdl was replaced by CPEc in the revised manuscript.
22- From the Bode-phase angle graphs, it seems that the inhibitor-containing samples have two time constants during immersion for 1 and 2 days.
Response to comment: Thank you for your comment. Yes, it seems that the sample with LDH-NO2- after immersion for 1 and 2 d presents two phase angle peaks on the Bode plots (Fig.8(i)), while only one impedance arc can be observed on the Nyquist plots (Fig.8(g)). The presence of two time constants in this initial immersion is a really interesting phenomenon, which remains unclear now. We will focus this phenomenon and further work on it to investigate the underlying mechanism in the near future.
23- The claims made in lines 349 to 352 about the surface micrographs are not provable from the images.
Response to comment: Thank you for your comment. There is something wrong in the previous sentence. “Fig.9(c2)” in the “According to the top-view morphology in Fig.9(a1-c1), almost no obvious aggregates can be found in the blank epoxy coating (Fig.9(a1)) and the epoxy coating with 2% LDH-NO2- (Fig.9(c1)), while some aggregates could be found on the surface of epoxy coating with 2% LDH (Fig.9(c2))” should be revised as Fig.9(b1). The addition of unmodified LDH powders in the epoxy coatings could indeed cause aggregates, which was in good accordance with our previous publications (Coatings 2022, 12, 1631). The reason why the modification of LDH by inhibitors could promote the dispersing state in the epoxy coatings will be further investigated in our future work.
24- Why is there no trace of chlorine in the EDS results of the scratch area?
Response to comment: Thank you for your comment. Yes, no trace of chloride in the EDS results of the scratched area, which may be attributed to the fact that the corrosion area was washed by running water rigorously and chlorides were removed due to its soluble property in aqueous solution. This phenomenon was similar to that reported by Yue Su et al. and no chloride can be detected in the corrosion products as well (Chemical Engineering Journal 426 (2021) 131269).
25- Line 398: alpha and gamma need to be corrected.
Response to comment: Thank you for your comment. alpha and gamma have been corrected in this line.
26- No reference has been mentioned for analyzing the results of the Raman test.
Response to comment: Thank you for your comment. Reference related to Raman test was inserted accordingly.
27- Why is there no trace of the released anion from the carrier in the Raman and EDX results? How can you prove the self-healing properties?
Response to comment: Thank you for your comment. Yes, no released nitrites can be detected in both Raman and EDX results of the corrosion product, the related reason can be analyzed as follows. On one hand, the released nitrites acted as a strong oxidizer and the oxidization reaction occurs according to the following equation in neutral or basic environment:
2Fe2+ + 2OH- + 2NO2- → 2NO↑+ γ-Fe2O3 + H2O (Corrosion Science 64 (2012) 105-114; Applied Surface Science (2015) 924-932)
As can be seen from the above equation, nitrites can be transformed to NO gas and thus it cannot be detected in the Raman and EDX results. On the other hand, the quantity of the remaining released nitrites was small and may probably disappear due to its soluble property in aqueous solution under rigorous flowing water washing.
Please note that the self-healing properties can be demonstrated through the LEIS results. According to Fig.13, for the coating with LDH-NO2-, almost no corrosion can be found at the immersion of 1 h, this attractive phenomenon can be ascribed to the healing effect of the release nitrites from LDH-NO2-. Passive film was formed on the metal substrate surface as nitrites could act a strong oxidizer and the corrosion site was healed.
28- Please explain how LDH can convert corrosion products from iron hydroxide to hematite? through what mechanism?
Response to comment: Thank you for your comment. In fact, it’s not LDH that convert corrosion product from iron hydroxide to hematite, to be specific, it’s the release nitrites promoted this transformation. The detailed reaction equation in neutral or basic environment was listed as follows:
2Fe2+ + 2OH- + 2NO2- → 2NO + γ-Fe2O3 + H2O (Corrosion Science 64 (2012) 105-114; Applied Surface Science (2015) 924-932)
Please note that we have added this reaction in the revised manuscript.
29- The conclusion section is incomplete. For example, there is no mention of the results of the EIS test in the solution phase and the results of the molybdate-containing sample.
Response to comment: Thank you for your comment. The results of the EIS test in the solution phase and the results of the molybdate-containing sample were added in the Conclusion part now.

Round 2
Reviewer 1 Report
The authors corrected the paper following the reviewer's advice and improved the quality of the manuscript. It can be accepted as it is.
Reviewer 3 Report
Thank you for the revision. The revised manuscript can be accepted.
Minor editing of English language required